# Daily Sugar-Sweetened Beverage Intake and Its Association with Undiagnosed Non-Communicable Diseases Among Malaysian Adults: Findings from a Nationally Representative Cross-Sectional Study

**DOI:** 10.3390/nu17101740

**Published:** 2025-05-20

**Authors:** Shi-Hui Cheng, Sumarni Mohd-Ghazali, Chee-Cheong Kee, Lay-Kim Tan

**Affiliations:** 1Faculty of Science and Engineering, School of Biosciences, University of Nottingham Malaysia, Semenyih 43500, Selangor, Malaysia; shihui.cheng@nottingham.edu.my; 2Biomedical Epidemiology Unit, Special Resource Centre, Institute for Medical Research, Ministry of Health Malaysia, Shah Alam 40170, Selangor, Malaysia; sumarni.mg@moh.gov.my; 3Sector for Biostatistics & Data Repository, Office of NIH Manager, National Institutes of Health, Ministry of Health Malaysia, Shah Alam 40170, Selangor, Malaysia; kee@moh.gov.my

**Keywords:** sugar-sweetened beverages, adults, hypertension, diabetes, hypercholesterolemia, obesity

## Abstract

**Background/objective:** We examined the prevalence of daily sugar-sweetened beverage (SSB) intake, identifying its sociodemographic determinants and exploring its potential association with undiagnosed non-communicable diseases (NCDs) among Malaysian adults. **Methods:** This cross-sectional study analyzed data from 6596 Malaysian adults participating in the 2019 National Health and Morbidity Survey (NHMS). Multiple logistic regression was used to examine the association between daily SSB intake and the risk of undiagnosed diabetes, hypertension, hypercholesterolemia, and obesity while adjusting for potential confounders. **Results:** The prevalence of daily SSB intake was 53.6%, with higher intake observed among females, older adults, Indians, and unemployed individuals. After adjusting for confounders, daily SSB intake was not significantly associated with undiagnosed diabetes (adjusted OR: 1.01, 95% CI: 0.80–1.29), undiagnosed hypertension (adjusted OR: 0.99, 95% CI: 0.81–1.22), undiagnosed hypercholesterolemia (adjusted OR: 0.99, 95% CI: 0.83–1.18), or obesity (adjusted OR: 1.08, 95% CI: 0.91–1.27). **Conclusions:** This study highlights the high prevalence of daily SSB intake among Malaysian adults, driven by sociodemographic factors. While a lack of direct associations with undiagnosed NCDs was observed, the high prevalence of SSB intake raises concerns about long-term health impacts. Targeted public health interventions are essential to address the cultural and economic determinants of SSB intake, as well as future research adopting longitudinal designs to explore how sustained reductions in SSB intake influence the risk of developing NCDs.

## 1. Introduction

Non-communicable diseases (NCDs), such as diabetes mellitus, hypertension, and hypercholesterolemia, are leading causes of illness and death worldwide. The World Health Organization (WHO) reports that NCDs account for over 70% of deaths globally, with low- and middle-income countries facing a disproportionate burden due to limited access to healthcare [1]. In Malaysia, the prevalence of NCDs has risen alarmingly in recent years, largely driven by lifestyle changes, rapid urbanization, and shifts in dietary habits [2].

Despite various public health initiatives, a substantial proportion of Malaysians with NCDs remain unaware of their status, hindering early intervention and effective management. The 2023 National Health and Morbidity Survey (NHMS) revealed that approximately two in five adults with diabetes, one in two adults with hypercholesterolemia, and 11.9% of adults with hypertension were unaware of their conditions [2]. This undiagnosed burden is particularly concerning, as these diseases may silently progress to severe complications such as cardiovascular disease, kidney failure, or stroke, and be detected only at these advanced stages [3].

Dietary factors are crucial in the development and progression of NCDs, with sugar-sweetened beverages (SSBs) identified as a significant risk factor. SSBs are defined as drinks that contain added sugars, including sodas, energy drinks, flavored milk, pre-mixed beverages, and sugar-enhanced traditional drinks [4]. These beverages are major contributors to the intake of free sugars in modern diets, providing excessive calories without essential nutrients. In Malaysia, the daily SSB intake is notably high, with the average daily sugar intake from SSBs being 59.14 ± 51.28 g, equivalent to about 12 teaspoons of sugar [5].

The rising intake of sugar-sweetened beverages (SSBs) is a significant public health concern due to its strong associations with obesity and NCDs. For example, high daily SSB intake has been linked to obesity, insulin resistance, elevated blood pressure, and dyslipidemia, all of which can lead to NCDs [6]. However, there is a lack of nationally representative data to contextualize these associations within Malaysia’s diverse dietary habits and socioeconomic factors. Addressing this knowledge gap is crucial to understanding the hidden prevalence of undiagnosed NCDs and their relationship with SSB intake. Therefore, this study aims to examine the association between daily SSB consumption and the prevalence of undiagnosed NCDs, utilizing data from the 2019 National Health and Morbidity Survey (NHMS). By focusing on undiagnosed cases, the research seeks to reveal crucial insights into the hidden burden of NCDs, which may pose a challenge to Malaysia’s public health system. The findings will provide essential evidence to inform policy development and enhance public health strategies tailored to the Malaysian context.

## 2. Materials and Methodology

### 2.1. Study Population

The present study utilized data from the 2019 National Health and Morbidity Survey (NHMS 2019). NHMS 2019 was a nationwide survey conducted by the Institutes for Public Health, National Institutes of Health, and Ministry of Health of Malaysia (MOH) between 14 July and 2 October 2019. The survey aimed to assess the prevalence of non-communicable diseases, risk factors for non-communicable diseases, healthcare demand, and health literacy in Malaysia. The survey covered both rural and urban areas across Malaysia. The detailed methodology of NHMS 2019, including the sampling design, survey materials, data collection forms, and codebook, is available in a technical report [7]. In this paper, only the key aspects of the NHMS 2019 survey methodology salient to this study are described.

### 2.2. Sampling Frame

The sampling frame was provided and sampling for NHMS 2019 was performed by the Department of Statistics Malaysia (DOSM). Briefly, geographical areas in Malaysia were divided into enumeration blocks (EBs) and further classified as urban or rural areas by DOSM based on the population of the gazetted areas. The gazetted areas with populations of 10,000 individuals or more were classified as urban EBs, whereas gazetted areas with populations of less than 10,000 were classified as rural EBs. Meanwhile, each EB contained between 80 and 120 living quarters (LQs), and each LQ housed between 500 and 600 individuals. The survey excluded institutional living quarters (LQs), e.g., institutional buildings such as hotels, hospitals, care homes, nursing homes, and holiday and recreation homes.

### 2.3. Sampling Design

The NHMS 2019 employed a two-stage stratified random sampling design, consisting of a primary stratum and a secondary stratum, to select respondents representative of the Malaysian population. The primary stratum included all thirteen states and three federal territories in Malaysia, whereas the secondary stratum consisted of the urban and rural strata within the primary stratum. In the first stage, the primary sampling units, i.e., EBs, were randomly selected from each sub-stratum. Secondary sampling units, which were the living quarters (LQs), were identified within the selected EBs. Based on the 2010 Population and Housing Census conducted by the Department of Statistics Malaysia, there were more than 75,000 EBs that were identified, and a total of 475 EBs were selected (362 urban EBs and 113 rural EBs) for the NHMS 2019 study. A total of 5676 LQs were selected from the identified 475 EBs. The second stage involved a random sampling of an average of 12 living quarters (LQs) within each of the selected EBs. All occupants of the selected LQs who had resided there at least two weeks before data collection were eligible to participate in the survey.

### 2.4. Survey Respondents

Our criteria for eligibility in this study were Malaysian citizens, aged 18 years or older, with no previous clinical diagnosis of diabetes mellitus, hypertension, and/or hypercholesterolemia. A total of 14,965 NHMS 2019 survey participants completed the NCD component of the survey. Before statistical analysis, the dataset consisted of 14,965 respondents who were screened with the following criteria. First, individuals who were non-Malaysians/citizenship unknown (*n* = 901) and aged below 18 years old (*n* = 4304) were excluded, leaving 9760 respondents in the dataset. Next, respondents who had been clinically diagnosed with diabetes mellitus, hypertension, and/or hypercholesterolemia (*n* = 3164) were removed. The final dataset, comprising data from 6596 respondents, was used for further analyses.

### 2.5. Ethical Consideration

The NHMS 2019 was officially registered with the National Medical Research Register (NMRR), MOH, and bore the registration number NMRR-18-3085-44207. The study obtained ethical clearance from the Medical Research and Ethics Committee (MREC) of MOH prior to the commencement of the survey. All identified eligible participants were informed about the survey via the participant’s information sheet, and informed written consent was obtained from individuals who agreed to participate before conducting the survey interview and related assessments.

### 2.6. Survey Materials and Data Collection

A structured questionnaire was used to collect data based on the survey scopes. The questionnaire was available in two formats: programmed in an application or in paper form. The participants were allowed to choose the mode of data collection: face-to-face interviews or self-administered interviews. Should a participant choose face-to-face interviews, a trained data collection enumerator would use the bilingual questionnaire (in Malay and English languages) and be accompanied by a questionnaire manual as a guide to collect data. Otherwise, a multi-lingual, self-administered questionnaire in Malay, English, Chinese, or Tamil was provided to those who chose to complete the survey independently.

Trained nurses conducted clinical assessments of each respondent’s anthropometric measurements including weight, height, and waist circumference [8]. Calibrated scales (Tanita personal scale HD 319 and SECA stadiometer 213) were used for measuring weight and height. The weighing scales were calibrated with standard weights prior to use in the survey.

### 2.7. Study Variables

#### 2.7.1. Independent Variable

Daily SSB intake was defined as consuming at least one of the following different types of sugar-added beverages daily: (i) sugar-added self-prepared drinks (SASD), (ii) commercially packed ready-to-drink (CPRD) (carbonated and non-carbonated) beverages, and/or (iii) pre-mixed (PM) beverages. SASD was defined as preparations of coffee, tea, chocolate, and/or malted beverages that were added with sugar, sweetened condensed milk, and/or sweetened creamer. CPRD beverages were defined as carbonated drinks (e.g., cola, soda) and non-carbonated drinks (e.g., soy milk, chrysanthemum tea, lemon tea, and chocolate drink). Meanwhile, PM beverages were defined as instant drink products containing sugar and/or creamer (e.g., premix coffee, tea, chocolate, soy, and cereal). Respondents were asked to quantify the frequency and quantity of intake of each type of SSB. Two questions were developed to assess the intake of SASD (frequency per week, and quantity per intake), and three questions each for CPRD and PM beverages (frequency per week and quantity per intake).

Two questions were used to assess the intake of SASD (frequency per week and quantity per intake): Q1: “In a typical week, how many days did you drink coffee/tea with added sugar and/or sweetened condensed milk and/or sweetened creamer? Usually on the day that you drank coffee/tea with added sugar and/or sweetened creamer, how many cups did you take in a day?” Q2: “In a typical week, how many days did you drink chocolate and/or malted beverages with added sugar and/or sweetened condensed milk and/or sweetened creamer? Usually on the day that you drank chocolate and/or malted beverages with added sugar and/or sweetened creamer, how many cups did you take in a day?” Additionally, three questions were used to assess the CPRD and PM beverage frequency per week, quantity per intake, and quantity of sweetener (sugar/honey/sweetened milk or creamer): Q3: “In a typical week, how many days did you drink pre-mixed drinks (2-in-1, 3-in-1, and 4-in-1)? Usually on the day that you drank pre-mixed drinks (2-in-1, 3-in-1, and 4-in-1), how many sachets did you take?” Q4: “In a typical week, how many days did you drink carbonated drinks (in pack, can or bottle)? Usually on the day that you drank carbonated drinks, how many glasses did you take?” Q5: “In a typical week, how many days did you drink non-carbonated drinks (in pack, can or bottle)? Usually on the day that you drank non-carbonated drinks, how many glasses did you take?”

#### 2.7.2. Dependent Variables

Respondents were subjected to blood sugar, cholesterol, and blood pressure measurements, and asked whether they had been clinically diagnosed with diabetes mellitus, hypertension, or hypercholesterolemia by a registered healthcare professional within the past twelve months. Respondents with fasting capillary blood glucose levels of 6.1 mmol/L or more (or non-fasting blood glucose of more than 11.1 mmol/L) and who had not been clinically diagnosed with diabetes mellitus in the past twelve months were classified as undiagnosed diabetes mellitus [9]. Respondents without a known history of hypertension but with a systolic blood pressure of 140 mmHg or higher and/or diastolic blood pressure of 90 mmHg or higher at enrollment were classified as having undiagnosed hypertension [10]. Respondents who were not previously clinically diagnosed with hypercholesterolemia but had blood cholesterol levels of 5.2 mmol/L or higher were classified as undiagnosed hypercholesterolemia [11]. All respondents with undiagnosed diabetes, undiagnosed hypertension, and/or undiagnosed hypercholesterolemia had self-reported never having received treatment for diabetes, hypertension, and/or hypercholesterolemia in the past twelve months.

## 3. Covariates

Several covariates were identified as potential confounders, including gender, age, ethnicity, residential area, marital status, educational levels, occupation status, monthly household income, and smoking status, and were included in the regression model. The age of the respondents was classified into five groups as follows: (i) 18–29 years, (ii) 30–39 years, (iii) 40–49 years, (iv) 50–59 years, and (v) 60 years and above. The respondents self-reported their ethnicity and were classified as follows: (i) Malay, (ii) Chinese, (iii) Indian, (iv) Bumiputera, and (v) Others. Residential areas were classified as urban or rural based on the EB in which it was located. Marital statuses were categorized as follows: (i) single, (ii) married, and (iii) widow/widower/divorcee. Education level was categorized based on the local Malaysian education system into (i) no formal education, (ii) primary education, (iii) secondary education, and (iv) tertiary education (all formal education received beyond secondary school).

Respondents’ total monthly household income was further categorized based on the Malaysian household income classification published by the Department of Statistics Malaysia, which is bottom 40% (B40) for monthly household income below RM 4850 (below USD1035), middle 40% (M40) if between RM 4850 and RM 10,959 (USD1035–2338), and top 20% (T20) if RM 10,960 and above (above USD2338). The smoking status of respondents was categorized as never, current, or past smokers. To categorize the status of adequate daily fruit and vegetable (FV) intake, we referred to the Malaysian Dietitian Guidelines 2020 [12]. Based on the guidelines, an individual was classified as having adequate daily FV intake when ≥5 servings of fruit and vegetables (e.g., ≥2 servings of fruit and ≥3 servings of vegetables) were consumed daily; otherwise, it was inadequate. The short version of the International Physical Activity Questionnaire was used to assess the level of physical activity of the respondents, where the physical activity was categorized as (i) inactive and (ii) active. The anthropometric measurements of the respondents were used to calculate the body mass index (BMI) by dividing the body weight (kg) by square height (m^2^). Respondents with a BMI of 25 and above were classified as overweight/obese, whereas the classification of abdominal obesity was having a waist circumference of 102 cm and above for men, and 88 cm and above for women [13].

### Statistical Analyses

All statistical analyses in the present study were conducted using the R programming language. The R packages “tidyverse”, “labelled”, “summarytools”, “tibble”, “survey”, “gtsummary”, and “flextable” were used in RStudio software [version 4.3.1] to carry out the descriptive statistics and multivariable logistic regression analyses for complex samples [14]. Additionally, classification tables and receiver operating characteristic (ROC) analyses were carried out to evaluate the predictive ability of the final model.

Prior to analysis, a structured representation of the survey data, i.e., a survey design object, was created by incorporating the sampling weights, stratification, and clustering. The survey design object was subsequently applied in all the analyses for complex samples. Descriptive analysis was conducted to determine the proportion of respondents by sociodemographic characteristics such as gender, age, ethnicity, residential area, marital status, education level, occupational status, monthly household income, as well as lifestyle risk factors (e.g., smoking, adequate daily FV intake, physical activity). Following that, Pearson’s chi-squared was performed to assess associations between sociodemographic characteristics, lifestyle factors, and daily SSB intake. Prevalence and 95% confidence intervals (CI) of general obesity, abdominal obesity, undiagnosed diabetes mellitus, undiagnosed hypertension, undiagnosed hypercholesterolemia, and daily SSB intake were illustrated in graphical charts. Unweighted sample sizes (n) were reported alongside weighted population-level prevalence estimates (%). Complex-sample multivariable logistic regression was conducted to examine the association between daily SSB intake and NCDs, and results were reported as adjusted odds ratios (aORs) with their respective 95% CIs, with *p*-values less than 0.05 interpreted as statistically significant. The R statistical packages “ggplot2” and “forestplot” were used to generate the charts.

## 4. Results

Table 1 presents the weighted distribution of sociodemographic and health characteristics among Malaysian adults aged 18 years and above, based on data from NHMS 2019. Although the unweighted sample size was 6596, all percentages reflect weighted population estimates. Our data showed a slightly higher proportion of male respondents (51.0%). The majority of the respondents were in the age group between 18 and 29 years old (38.5%), of Malay origins (57.1%), living in urban areas (78.7%), and married (59.5%). Around 52.8% of the respondents completed their education at the secondary level, 68.1% were employed (employed/self-employed), and 62.2% had a monthly household income below USD1035. Approximately 23% of the respondents were current smokers and 4% of them were past smokers. An alarmingly low prevalence of daily adequate FV intake was observed at 2.3% (95% CI: 1.81–2.90). One-third of the Malaysian adults were physically active (75.7%, 95% CI: 74.1–77.3). More than half of the respondents (53.6%) reported having at least one SSB intake daily. Estimation of the prevalence of NCDs in the study population demonstrated that general obesity had the highest prevalence (63.1%, 95% CI: 8.1–10.40), followed by abdominal obesity (48.0%, 95% CI: 46.0–50.1), undiagnosed hypercholesterolemia (29.5%, 95% CI: 27.5–31.7), undiagnosed hypertension (16.9%, 95% CI: 15.6–18.2), and undiagnosed diabetes mellitus (9.2%, 95% CI: 8.1–10.4) (Figure 1).

Table 2 presents the univariable analysis. Our data demonstrated that daily SSB intake was significantly higher among females (50.2%) compared to males (42.7%) (*p* < 0.001) (Table 2). Prevalence of daily SSB intake increased significantly with age, with the oldest age group (above 60 years old) having the highest prevalence (62.6%) (*p* < 0.001). Respondents of Indian origin had the highest prevalence of daily SSB intake (51.9%) compared to Bumiputera (28.3%) and other ethnic groups (28.5%) (*p* < 0.001), which was statistically significant. Respondents who were single had the lowest prevalence of SSB intake (35.4%) when compared to those who were married (52.5%) or widowed(er)/divorced (54.0%) (*p* < 0.001). The majority of unemployed respondents and non-smokers had daily SSB intake, with prevalences of 51.3% and 49.1%, respectively. It is important to note that these findings represent associations rather than causal relationships and may reflect broader behavioral, cultural, or socioeconomic patterns influencing dietary choices in the Malaysian adult population.

We observed a higher prevalence of daily SSB intake among undiagnosed diabetes mellitus (51%), undiagnosed hypercholesterolemia (50.4%), undiagnosed hypertension (49.8%), and abdominal obesity (47.6%) when compared to those who are non-diagnosed (Figure 2). The daily SSB intake prevalence was comparable between those who were overweight/obese (46.9%) and not obese (46.1%).

The association analyses revealed no significant association between daily SSB intake with undiagnosed diabetes mellitus (aOR: 1.04, 95% CI: 0.82–1.32), undiagnosed hypertension (aOR: 1.03, 95% CI: 0.85–1.25), undiagnosed hypercholesterolemia (aOR: 1.01, 95% CI: 0.85–1.20), general obesity (aOR: 0.82, 95% CI: 0.61–1.10), and abdominal obesity (aOR: 1.08, 95% CI: 0.92–1.28) when sociodemographic characteristics and lifestyle factors were held constant (Figure 3). There were no significant interactions among the independent variables. The model prediction values of the final model for undiagnosed diabetes mellitus, undiagnosed hypertension, undiagnosed hypercholesterolemia, general obesity, and abdominal obesity were the respective 90.6% (area under curve (AUC): 0.69, 95% CI: 0.67–0.71, *p* < 0.0001), 83.7% (AUC: 0.73, 95% CI: 0.71–0.74, *p* < 0.0001), 72.3% (AUC: 0.69, 95% CI: 0.67–0.70, *p* < 0.0001), 63.3% (AUC: 0.59, 95% CI: 0.57–0.61, *p* < 0.0001), and 56.8% (AUC: 0.68, 95% CI: 0.67–0.70, *p* < 0.0001), respectively. The predictive models demonstrated moderate accuracy in identifying undiagnosed NCDs, particularly for undiagnosed hypertension (AUC: 0.73) and diabetes mellitus (AUC: 0.69), indicating acceptable discriminative ability. The models for undiagnosed hypercholesterolemia (AUC: 0.69) and abdominal obesity (AUC: 0.68) also showed fair performance. However, the model for general obesity had limited predictive power (AUC: 0.59), suggesting reduced discrimination for this outcome. These findings suggest that the selected independent variables have reasonable utility for predicting undiagnosed NCDs, particularly for conditions with clearer clinical or behavioral risk patterns.

## 5. Discussion

The study demonstrates the high prevalence of daily SSB intake among Malaysian adults and its significant public health implications. SSBs are drinks that contain added sugars, such as sucrose, high-fructose corn syrup, or fruit juice concentrates. These added sugars contribute significantly to the high energy content of SSBs while offering minimal nutritional value. The observed trend is consistent with findings from the Malaysian Adults Nutrition Survey, which reported that 55.9% of Malaysian adults consumed SSB daily [15]. The persistent high intake of SSB over the years suggests that existing public health interventions may not have been sufficiently effective in curbing high daily SSB intake. Due to the burden of health-related consequences, these findings underscore the pressing need for more targeted and culturally tailored public health strategies. Similar findings were reported by Cheng and Lau (2022), who observed that sugar consumption per day in Malaysian adults was around 12 teaspoons [5]. This trend mirrors that of other Southeast Asian countries, such as Thailand [16] and Indonesia [17], where rapid urbanization and increased availability of processed beverages have contributed to SSBs’ popularity.

Notably, this study found a higher prevalence of daily SSB consumption among females and older adults. Given that SSBs are a major source of empty calories, higher intake among females may contribute to the greater prevalence of overweight and obesity observed in Malaysian women compared to men, according to NHMS 2019 [18]. This aligns with findings indicating that Malaysian older adults consume an average of eight teaspoons per day [19]. Furthermore, older adults often prefer sweetened beverages such as tea and coffee prepared with added sugar and sweetened condensed milk, as well as traditional *kuih* (local desserts). These culturally embedded food and drink contribute significantly to their overall sugar intake [19]. This pattern mirrors findings from Singapore, where 90.5% of the elderly population was reported to exceed the recommended daily sugar intake [20]. Financial constraints among the elderly, often due to retirement and rising living costs, can further compromise diet quality. Additionally, the preference for SSBs over healthier alternatives such as milk reduces calcium intake, which is especially concerning in the elderly due to their increased risk of osteoporosis and falls [21]. To address the increasing prevalence of daily SSB intake among Malaysian adults, the local public health authorities should enhance strategies that promote healthier beverage consumption. The recent introduction of tax relief for low-sugar beverages in Malaysia represents a positive fiscal measure to incentivize healthier choices among consumers [22]. To maximize its impact, this policy should be complemented with robust public education campaigns and structural interventions, including clear front-of-package labeling and visual indicators of sugar content to support informed decision making [23]. Furthermore, reinforcing dietary literacy through the dissemination of the Malaysian Food Pyramid and its emphasis on reduced sugar intake remains crucial [12]. Effective outcomes require strong cross-sector collaboration involving schools, healthcare providers, and the food and beverage industry to ensure the delivery of consistent health messages and support long-term behavior change. A coordinated strategy that integrates fiscal measures, public education, and regulatory interventions offers a comprehensive approach to reducing sugar consumption and enhancing population health.

The finding that individuals of Indian ethnicity had higher SSB intake emphasizes the influence of cultural factors on dietary behaviors. Traditional Indian foods such as *roti canai* (a type of flatbread) and sweetened beverages like *teh tarik* (frothy milk tea) are not only culturally significant but also widely available and inexpensive, making them more prevalent within this group. This is in line with findings from studies in multicultural societies, where dietary patterns are strongly related to cultural identity and access to affordable food and beverages [24]. Furthermore, economic accessibility plays a crucial role in influencing dietary choices. In Malaysia, street food vendors and food courts offer SSB at reasonable prices. This affordability, coupled with the cultural integration of these foods and beverages into daily routines, contributes to increased SSB consumption among Indians. Previous research has shown that Indians have the highest median HbA1c levels (8.3%), followed by Malays (7.7%) and Chinese (7.2%) [25], indicating a greater risk of diabetes and other metabolic disorders. This elevated risk may, in part, be attributed to the increased consumption of SSB among the Indian population [26].

Our findings also indicate that single individuals had a lower likelihood of daily SSB intake (35.4%) compared to married (52.5%) and widowed/divorced individuals (54.0%). This aligns with previous research showing that married individuals tend to consume fast food more frequently, often more than three times a week, where SSBs are typically included as part of the set meals or served with soft drinks, thereby increasing the overall intake of SSBs [27]. Additionally, marriage has been linked to an increased probability of overweight and obesity [28], likely due to changes in eating habits and parenting stresses [29]. Parents may find meal preparation at home particularly difficult due to time constraints and a hectic home environment [29]. As a result, fast food restaurants become a more convenient dining option, where SSBs are readily available and often included as standard menu items. This dining pattern could result in higher SSB consumption for both parents and children, further contributing to the family’s overall sugar intake. In addition, emotional eating and food-related coping mechanisms in the context of marital stress may further promote the consumption of high-calorie, sugary beverages [30]. Our findings also align with research from Thailand, which reported that individuals who experience loneliness, particularly widowed or divorced individuals, had a higher likelihood of purchasing sugary food and beverages at convenience stores [31]. This suggests that emotional and social factors may influence dietary choices across different marital statuses and highlights the need for targeted interventions that address emotional well-being and improve access to healthier beverage options.

Unemployed individuals in the present study exhibited a higher prevalence of daily SSB consumption, consistent with previous studies that associate higher SSB intake with lower income levels [32,33]. Financial constraints often limit access to healthier food options, making fast food and SSB more attractive as affordable and convenient meal choices [32]. However, these low-cost dietary patterns are typically high in calories and low in essential nutrients, increasing the risk of nutritional deficiencies and non-communicable diseases such as diabetes, obesity, and cardiovascular conditions [34]. This aligns with findings from the United States National Health and Nutrition Examination Survey (NHANES), which reported that individuals from low-income and food-insecure households had significantly higher consumption of ultra-processed foods and SSBs [35]. High SSB consumption among lower-income and unemployed populations necessitates targeted interventions focused on accessibility and affordability. Empowering these groups to prepare more home-cooked meals is a key strategy, as home-cooked meals are often more affordable compared to fast foods and sugary drinks and are also associated with better dietary quality [36]. To complement this, strategic subsidies on healthy beverage options, such as fresh fruit juices or milk, alongside a progressive sugar tax on SSBs, could further reduce excessive sugar consumption [37]. Additionally, initiatives that enhance nutritional literacy among low-income populations could foster long-term behavioral changes and improve dietary quality among the low-income populations [38].

Interestingly, Malaysian non-smokers were more likely to consume SSBs daily (49.1%), which may be attributed to differing health behavior clusters—where smokers may be more inclined to consume caffeine or alcohol instead of sweetened beverages or may prioritize other substances over sugar intake, which is pending further investigation in our population [39,40]. These further highlight the importance of future studies to investigate the complex interplay between social, behavioral, and economic factors in shaping dietary practices and underscore the importance of considering contextual and population-level influences when designing public health interventions targeting SSB reduction in our population.

Overall, our study found significant associations between daily SSB intake and several sociodemographic and behavioral factors. Though these associations may appear modest, they are statistically meaningful at the population level due to the large, nationally representative sample and the application of complex survey weights. The higher prevalence of daily SSB intake among married individuals and those currently employed may reflect lifestyle routines, social eating patterns, or greater household purchasing power, as supported by previous studies showing similar trends in SSB consumption among different household compositions and income levels [41,42].

Despite the well-established global association between SSB consumption and NCDs [6], our study did not find a significant association between daily SSB intake and undiagnosed NCDs, after adjusting for sociodemographic and lifestyle factors. These findings diverge from previous research where excessive SSB intake has been consistently linked to obesity [43] and NCDs [6]. A possible explanation lies in the confounding effects of other factors such as dietary patterns, physical activity levels, and genetic predisposition may play a crucial role in the development of NCDs. This aligns with previous research indicating that multiple dietary habits, physical activity, and lifestyle factors contribute to NCDs [44]. Secondly, the cross-sectional nature of this study restricts the ability to draw conclusions about causality. The absence of an association does not imply that SSB has no impact on undiagnosed NCDs but rather suggests that within this dataset, a direct link was not observed. Longitudinal studies with detailed dietary assessments such as food frequency questionnaires and 24-h dietary recall may be better suited to capturing the long-term effects of habitual SSB consumption on disease development. Furthermore, the absence of a significant association does not equate to the absence of risk. Our study highlights key sociodemographic groups with higher SSB consumption (e.g., females, older adults, Indians, unemployed individuals, and non-smokers), which is valuable for public health targeting. Identifying these patterns supports the development of targeted and culturally relevant interventions. In line with the existing literature and global evidence consistently linking high SSB intake to NCD risk, our findings further reinforce the need for continued monitoring, awareness, and prevention strategies, especially in vulnerable subpopulations.

Nevertheless, the high prevalence of SSB intake among individuals with undiagnosed NCDs in this study raises concerns about its potential role in exacerbating existing health conditions. SSBs are known to contribute to increased caloric intake, insulin resistance, and inflammation, which can worsen the metabolic profiles of individuals with undiagnosed NCDs [45]. Similarly, data from the China Health and Nutrition Survey (CHNS) [46] and the Korean National Health and Nutrition Examination Survey [47] indicated that higher free sugar intake, predominantly from SSBs, was associated with an increased risk of obesity and metabolic syndrome. This underscores the importance of early detection of NCDs and concurrent reduction in SSB consumption as part of integrated public health strategies.

One of the key strengths of this study is its large, nationally representative sample, which enhances the generalizability of the findings to the broader population. Another strength lies in the study’s focus on undiagnosed metabolic conditions, providing valuable insights into the hidden burden of diseases that could otherwise go undetected in routine health screenings. Moreover, the study contributes valuable insights into sociodemographic disparities in SSB consumption, providing an evidence base for tailored public health interventions. By identifying at-risk groups such as females, Indians, older adults, and those from lower socioeconomic backgrounds, this research offers practical recommendations for policy makers and public health professionals.

Nevertheless, several limitations must be acknowledged. The cross-sectional design hinders the establishment of causal relationships between daily SSB consumption and undiagnosed NCDs, as it only captures associations at a single point in time. Additionally, residual confounding factors may still exist even after confounding variables have been adjusted for, particularly from unmeasured factors such as total energy intake, overall dietary patterns, and levels of physical activity. Another limitation is the use of self-reported dietary intake, which may be subject to recall bias and underestimation of actual SSB consumption. Additionally, we did not quantify the types or volumes of SSBs consumed; therefore, we were unable to estimate their specific caloric contributions. Future research should incorporate longitudinal study designs and incorporate comprehensive dietary assessments, including total energy intake and beverage types to obtain a better understanding of these associations.

## 6. Conclusions

This study highlights the high prevalence of daily SSB intake among adults in Malaysia, particularly females, older adults, those of Indian descent, those who are unemployed, those who are widowed/divorced, and non-smokers. While no direct association was found between daily SSB intake and undiagnosed NCDs, the high prevalence of SSB intake among individuals in these demographic groups raises concerns about its potential role in exacerbating health outcomes. These findings emphasize the urgent need for multifaceted public health interventions that not only target SSB reduction but also address the broader lifestyle and sociodemographic determinants. Future research should focus on longitudinal studies to better understand the long-term impact of dietary and lifestyle modifications in mitigating NCD prevalence and improving public health outcomes in Malaysia.

## Figures and Tables

**Figure 1 nutrients-17-01740-f001:**
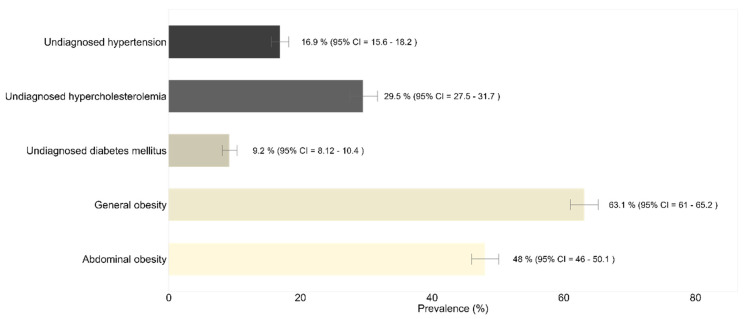
Prevalence of abdominal obesity, general obesity, undiagnosed diabetes mellitus, undiagnosed hypertension, and undiagnosed hypercholesterolemia among Malaysian adults aged 18 years and above (*n* = 6596).

**Figure 2 nutrients-17-01740-f002:**
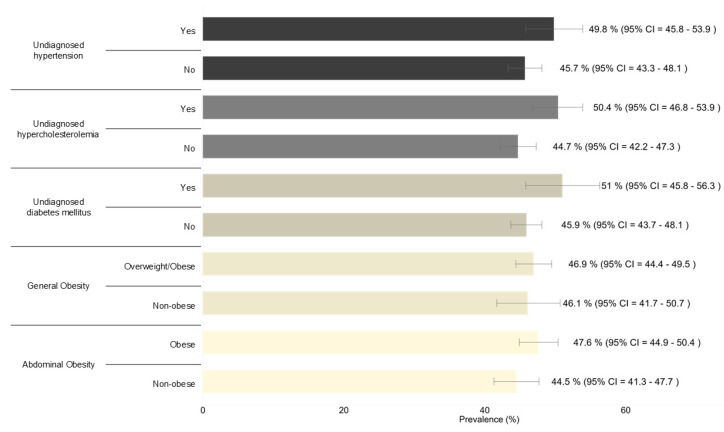
Prevalence of daily SSB intake among Malaysian adults aged 18 years and above with abdominal obesity, general obesity, undiagnosed diabetes mellitus, undiagnosed hypertension, and undiagnosed hypercholesterolemia (*n* = 6596).

**Figure 3 nutrients-17-01740-f003:**
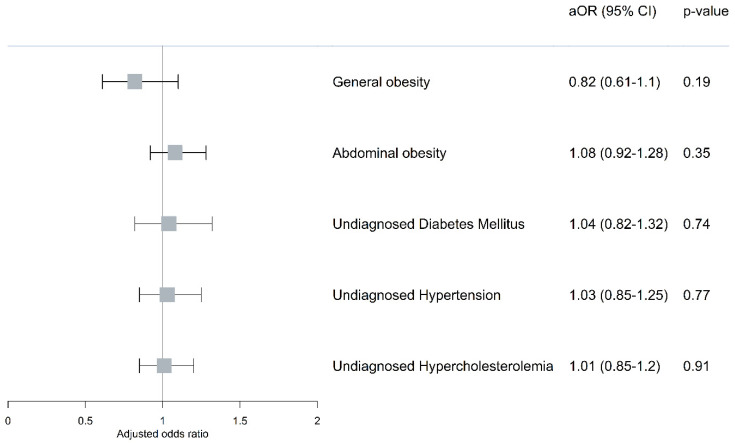
Associations between daily SSB intake with general obesity, abdominal obesity, diabetes mellitus, hypertension, and hypercholesterolemia among Malaysian adults aged 18 years and above. Multiple logistic regression was performed and adjusted for sociodemographic (e.g., gender, age, ethnicity, residential area, marital status, education, occupational status, and monthly household income) and lifestyle risk factors (e.g., smoking status, adequate daily FV intake, and physical activity). aOR: adjusted odds ratio; 95% CI: 95% confidence interval.

**Table 1 nutrients-17-01740-t001:** Weighted distribution of sociodemographic and health characteristics among Malaysian adults aged 18 years and above (NHMS 2019, *n* = 6596).

Characteristic	Estimated Population	Count (*n*)	Percentage (%)	95% CI ^1^
Gender				
Male	7,236,316	3083	51.0	49.6–52.5
Female	6,942,798	3513	49.0	47.5–50.4
Age group (years old)				
18–29	5,460,514	2008	38.5	36.3–40.8
30–39	3,799,665	1627	26.8	24.8–28.9
40–49	2,312,903	1167	16.3	15.0–17.7
50–59	1,558,223	971	11.0	9.97–12.1
>60	1,047,808	823	7.4	6.58–8.30
Ethnicity				
Malay	8,090,716	4547	57.1	52.6–61.4
Chinese	3,390,926	850	23.9	19.8–28.6
Indian	816,321	390	5.8	4.48–7.38
Bumiputera	1,766,091	765	12.5	10.4–14.8
Others	115,059	44	0.8	0.44–1.48
Residential area				
Urban	11,156,093	4062	78.7	77.0–80.3
Rural	3,023,021	2534	21.3	19.7–23.0
Marital status				
Single	5,129,538	1914	36.2	33.9–38.6
Married	8,430,747	4239	59.5	57.0–61.9
Widow(er)/Divorcee	618,829	443	4.4	3.78–5.04
Education level				
No formal education	302,693	214	2.1	1.72–2.66
Primary education	1,800,633	1070	12.7	11.6–14.0
Secondary education	7,463,394	3386	52.8	50.6–55.0
Tertiary education	4,561,585	1903	32.3	30.0–34.7
Occupation status				
Yes	9,659,412	4252	68.1	66.2–70.0
No	4,518,571	2343	31.9	30.0–33.8
Monthly household income				
Bottom 40%	8,401,239	4110	62.2	59.1–65.2
Middle 40%	3,820,170	1647	28.3	25.6–31.1
Top 20%	1,287,802	537	9.5	7.61–11.9
Smoking				
Never	10,383,579	4878	73.2	71.4–75.0
Current	3,226,601	1422	22.8	21.1–24.5
Past	568,933	296	4.0	3.40–4.73
Adequate daily FV intake				
Adequate	324,986	130	2.3	1.81–2.90
Inadequate	13,843,103	6458	97.7	97.1–98.2
Physical activity				
Inactive	3,399,399	1587	24.3	22.7–25.9
Active	10,588,833	4929	75.7	74.1–77.3
Daily SSB intake				
No	6,572,784	3180	46.4	44.3–48.5
Yes	7,595,306	3408	53.6	51.5–55.7

^1^ 95% CI: 95% confidence interval. All percentages are weighted to reflect population-level estimates from the National Health and Morbidity Survey (NHMS) 2019. Frequencies (n) are unweighted counts; percentages (%) are weighted and may not sum to exactly 100% due to rounding.

**Table 2 nutrients-17-01740-t002:** Daily sugar-sweetened beverage (SSB) intake by sociodemographic factors among NHMS 2019 respondents, Malaysian adults aged 18 years and above (*n* = 6596).

			Daily SSB Intake (*n* = 3408)		No SSB Intake (*n* = 3180)		
Characteristic	Estimated Population	Count (*n*)	Prevalence (%)	95% CI ^1^	Prevalence (%)	95% CI ^1^	*p*-Value
Gender							<0.001
Male	7,236,316	3083	42.7	39.8–45.7	57.3	54.3–60.2	
Female	6,942,798	3513	50.2	47.6–52.9	49.8	47.1–52.4	
Age group (years old)							<0.001
18–29	5,460,514	2008	36.5	33.4–39.6	63.5	60.4–66.6	
30–39	3,799,665	1627	46.8	42.6–51.0	53.2	49.0–57.4	
40–49	2,312,903	1167	53.7	49.2–58.1	46.3	41.9–50.8	
50–59	1,558,223	971	58.6	54.2–62.9	41.4	37.1–45.8	
>60	1,047,808	823	62.6	57.3–67.5	37.4	32.5–42.7	
Ethnicity							<0.001
Malay	8,090,716	4547	49.3	46.9–51.6	50.7	48.4–53.1	
Chinese	3,390,926	850	48.3	42.5–54.0	51.7	46.0–57.5	
Indian	816,321	390	51.9	43.9–59.8	48.1	40.2–56.1	
Bumiputera	1,766,091	765	28.3	23.6–33.6	71.7	66.4–76.4	
Others	115,059	44	28.5	18.3–41.4	71.5	58.6–81.7	
Residential area							0.46
Urban	11,156,093	4062	46.1	43.5–48.6	53.9	51.4–56.5	
Rural	3,023,021	2534	47.6	44.5–50.7	52.4	49.3–55.5	
Marital status							<0.001
Single	5,129,538	1914	35.4	31.9–39.2	64.6	60.8–68.1	
Married	8,430,747	4239	52.5	49.8–55.1	47.5	44.9–50.2	
Widow(er)/Divorcee	618,829	443	54.0	46.5–61.3	46.0	38.7–53.5	
Education level							0.16
No formal education	302,693	214	53.3	43.8–62.6	46.7	37.4–56.2	
Primary education	1,800,633	1070	51.2	46.7–55.7	48.8	44.3–53.3	
Secondary education	7,463,394	3386	45.5	42.7–48.3	54.5	51.7–57.3	
Tertiary education	4,561,585	1903	45.6	41.4–49.9	54.4	50.1–58.6	
Occupation status							<0.001
Yes	9,659,412	4252	44.1	41.7–46.6	55.9	53.4–58.3	
No	4,518,571	2343	51.3	48.2–54.4	48.7	45.6–51.8	
Monthly household income							0.12
Bottom 40%	8,401,239	4110	47.2	44.6–49.8	52.8	50.2–55.4	
Middle 40%	3,820,170	1647	42.8	38.9–46.7	57.2	53.3–61.1	
Top 20%	1,287,802	537	43.7	37.5–50.1	56.3	49.9–62.5	
Smoking							<0.001
Never	10,383,579	4878	49.1	46.6–51.6	50.9	48.4–53.4	
Current	3,226,601	1422	39.1	34.9–43.4	60.9	56.6–65.1	
Past	568,933	296	37.8	30.0–46.2	62.2	53.8–70.0	
Adequate daily FV intake							0.24
Adequate	324,986	130	52.9	41.7–63.8	47.1	36.2–58.3	
Inadequate	13,843,103	6458	46.2	44.1–48.3	53.8	51.7–55.9	
Physical activity							0.98
Inactive	3,399,399	1587	46.6	43.0–50.2	53.4	49.8–57.0	
Active	10,588,833	4929	46.5	44.1–48.9	53.5	51.1–55.9	

^1^ 95% CI: 95% confidence interval.

## Data Availability

All data generated during this study are included in the published article. However, to ensure data protection, the data used in this study are not publicly accessible. They can be obtained from the Sector for Biostatistics and Data Repository, Office of the NIH Manager, National Institutes of Health Malaysia, upon reasonable request and with approval from the Director General of the Ministry of Health Malaysia.

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
