# Peer review of "Daily Sugar-Sweetened Beverage Intake and Its Association with Undiagnosed Non-Communicable Diseases Among Malaysian Adults: Findings from a Nationally Representative Cross-Sectional Study"

_nutrients, 2025, doi:10.3390/nu17101740_

Round 1
Reviewer 1 Report
Comments and Suggestions for Authors
The study titled "Daily Sugar-Sweetened Beverages Intake and its’ Association with
Undiagnosed Non-communicable Diseases among Malaysian Adults: Findings from a
Nationally Representative Cross-Sectional Study" is a cross-sectional study conducted among Malaysian adults. The secondary source of data has been analyzed to examine the association between daily SSB intake and the risk of undiagnosed diabetes, hypertension, hypercholesterolemia, and obesity.
Some issues identified in the article are as follows:
In Table 1, it does not make a sense that the prevalence of demographic variables like gender, age group, and ethnicity etc. For example, what does it mean that the prevalence of males is 51% and females is 49%.
In Table 2, it is shown that there is a significant association of daily SSB intake with marital status, occupation, smoking etc which is not convincing
The NHMS 2019 data is useful in showing the prevalence of non-communicable disease among the Malaysian population
The findings indicate that there is no association between daily SSB consumption and undiagnosed diabetes, hypertension, hypercholesterolemia, and obesity. The association of daily SSB intake was only observed with sex, age group, ethnicity, etc., which are non-modifiable factors. However, the study recommends in conclusion for public health intervention. These findings may not be of interest to readers.
Comments on the Quality of English LanguageMay be improved
Author Response
For Research Article: nutrients-3565763
Response to Reviewer 1’s comments
- Summary
We thank you for taking your time to review this manuscript. The comments/suggestions are relevant. Please find detailed responses below and the corresponding revisions/ corrections highlighted/ in track changes in the re-submitted files.
|
2. Questions for General Evaluation |
Reviewer’s Evaluation |
Response and Revisions |
|
Does the introduction provide sufficient background and include all relevant references? |
Can be improved |
We have revised the Introduction section, and the English language has been proofread and edited for clarity and accuracy. |
|
Is the research design appropriate? |
Must be improved |
We have revised the Methodology section, and the English language has been proofread and edited for clarity and accuracy. |
|
Are the methods adequately described? |
Can be improved |
We have revised the Methodology section, and the English language has been proofread and edited for clarity and accuracy. |
|
Are the results clearly presented? |
Must be improved |
We have revised the Results section, and the English language has been proofread and edited for clarity and accuracy. |
|
Are the conclusions supported by the results? |
Must be improved |
We have revised the Conclusion and the English language has been proofread and edited for clarity and accuracy. |
- Point-by-point response to Comments and Suggestions for Authors
The study titled "Daily Sugar-Sweetened Beverages Intake and its Association with
Undiagnosed Non-communicable Diseases among Malaysian Adults: Findings from a
Nationally Representative Cross-Sectional Study" is a cross-sectional study conducted among Malaysian adults. The secondary source of data has been analysed to examine the association between daily SSB intake and the risk of undiagnosed diabetes, hypertension, hypercholesterolemia, and obesity.
Some issues identified in the article are as follows:
Comment 1: In Table 1, it does not make a sense that the prevalence of demographic variables like gender, age group, and ethnicity etc. For example, what does it mean that the prevalence of males is 51% and females is 49%.
Response 1: Thank you for your comment. We acknowledge the confusion and would like to clarify that the values presented in Table 1 are based on weighted data from a population-based dataset. The purpose of applying weights is to ensure that our estimates are representative of the broader population from which the sample was drawn.
Therefore, the reported percentages for demographic variables (e.g., gender, age group, ethnicity) represent population-level prevalence estimates, not the raw counts or proportions from our study sample. For instance, a weighted prevalence of 51% for males and 49% for females indicates that, after applying population weights, males constitute 51% of the represented population, even if the sample proportions might differ slightly.
To improve clarity, we will revise the table title and footnotes to explicitly indicate that weighted estimates are being reported (Page 6, line 275-276 and Page 7, footnote). Additionally, we have also revised the methodology (Page 6, line 250-251) and results section (Page 6, line 257-260).
Comment 2: In Table 2, it is shown that there is a significant association of daily SSB intake with marital status, occupation, smoking etc which is not convincing
Response 2: Thank you for your comment. We understand your concern regarding the significant associations observed between daily SSB intake and variables such as marital status, occupation status, and smoking. We would like to clarify that these associations were derived from a population-based, nationally representative dataset (NHMS 2019) using complex survey weights to account for the stratified multistage sampling design. Furthermore, we performed univariable analysis in Table 2. The p-values reported in Table 2 are based on design-adjusted statistical tests, which take into account the weighting effects. Therefore, the observed associations—although they may seem modest—are statistically significant at the population level due to the large sample size and representative nature of the data. To strengthen the clarity of our findings, we will revise the Results section (Page 8, line 279, line 288-291) to emphasize that these are associations, not causal relationships, and add interpretative context where appropriate.
Comment 3: The NHMS 2019 data is useful in showing the prevalence of non-communicable disease among the Malaysian population
Response 3: We thank the reviewer for the acknowledgement
Comment 4: The findings indicate that there is no association between daily SSB consumption and undiagnosed diabetes, hypertension, hypercholesterolemia, and obesity. The association of daily SSB intake was only observed with sex, age group, ethnicity, etc., which are non-modifiable factors. However, the study recommends in conclusion for public health intervention. These findings may not be of interest to readers.
Response 4: Thank you for your thoughtful comments. We acknowledge that our study did not identify statistically significant associations between daily sugar-sweetened beverage (SSB) intake and undiagnosed diabetes, hypertension, hypercholesterolemia, and obesity. However, the absence of a significant association does not equate to the absence of risk. Our study highlights key sociodemographic groups with higher SSB consumption (e.g. females, older adults, Indian, unemployed individuals and smoking status), which is valuable for public health targeting. Identifying these patterns supports the development of targeted, culturally relevant interventions. Moreover, existing literature and global evidence consistently link high SSB intake to NCD risk, and our findings reinforce the need for continued monitoring, awareness, and prevention strategies, especially in vulnerable subpopulations. We have revised our discussion on page 14 (line 472-480).

Reviewer 2 Report
Comments and Suggestions for Authors
Thank you for submitting the manuscript “Daily Sugar-Sweetened Beverages Intake and its’ Association with Undiagnosed Non-communicable Diseases among Malaysian adults: Findings from a Nationally Representative Cross-Sectional Study” to Nutrients. Here are some suggestions for improving the manuscript:
- Please check throughout the text that abbreviations are defined the first time they appear, e.g. abstract.
- What was characterized as SSB in this study? It needs to be described in the M&M.
- The methodology needs to be described in more detail. Further explaining the criteria adopted for data collection and analysis can increase the transparency and reproducibility of the study.
- The results presented could be better organized and explained. Some sections lack a clear connection between the data presented and the conclusions drawn. It is important to reinforce the explanation of how the findings relate to the objective of the study.
- The discussion section does not sufficiently explore the relationship between the study findings and the existing literature. It is necessary to include more relevant references and compare the results obtained with similar studies to strengthen the validity of the conclusions.
Author Response
For Research Article: nutrients-3565763
Response to Reviewer 2’s comments
- Summary
We thank you for taking your time to review this manuscript. The comments/suggestions are relevant. Please find detailed responses below and the corresponding revisions/ corrections highlighted/ in track changes in the re-submitted files.
|
2. Questions for General Evaluation |
Reviewer’s Evaluation |
Response and Revisions |
|
Does the introduction provide sufficient background and include all relevant references? |
Can be improved |
We have revised the Introduction section, and the English language has been proofread and edited for clarity and accuracy. |
|
Is the research design appropriate? |
Yes |
We thank the reviewer for the acknowledgement |
|
Are the methods adequately described? |
Can be improved |
We have revised the Methodology section, and the English language has been proofread and edited for clarity and accuracy. |
|
Are the results clearly presented? |
Can be improved |
We have revised the Results section, and the English language has been proofread and edited for clarity and accuracy. |
|
Are the conclusions supported by the results? |
Yes |
We thank the reviewer for the acknowledgement |
- Point-by-point response to Comments and Suggestions for Authors
Thank you for submitting the manuscript “Daily Sugar-Sweetened Beverages Intake and its’ Association with Undiagnosed Non-communicable Diseases among Malaysian adults: Findings from a Nationally Representative Cross-Sectional Study” to Nutrients. Here are some suggestions for improving the manuscript:
Comment 1: Please check throughout the text that abbreviations are defined the first time they appear, e.g. abstract.
Response 1: Thank you for your suggestion, we have defined the abbreviation of SSB in abstract
Comment 2: What was characterized as SSB in this study? It needs to be described in the M&M.
Response 2: In the methodology section under subheading “Independent variables”, we have defined SSB as taking at least one of the following different types of sugar added beverages daily: (i) sugar added self-prepared drinks (SASD), (ii) commercially packed ready-to-drink (CPRD) (i.e. carbonated and non-carbonated) and/or (iii) pre-mixed (PM) beverages. We further defined SASD as preparations of coffee, tea, chocolate, and/or malted beverages that were added with sugar, sweetened condensed milk and/or sweetened creamer. Meanwhile, CPRD beverages were defined as carbonated drinks (e.g. cola, soda) and non-carbonated drinks (e.g. soy milk, chrysanthemum tea, lemon tea, and chocolate drink), whilst PM was defined as instant drink products containing sugar and/or creamer (e.g. premix coffee, tea, chocolate, soy, and cereal). We have further revised the text under subheading “Independent variable” (Page 4, line 167-183) to provide more clarity to the characterization of SSB.
Comment 3: The methodology needs to be described in more detail. Further explaining the criteria adopted for data collection and analysis can increase the transparency and reproducibility of the study.
Response 3: We thank the reviewer for the valuable suggestion. As this study is a sub-analysis of the National Health and Morbidity Survey (NHMS) 2019 data, we have followed the survey’s established methodology. Briefly, the NHMS 2019 employed a nationally representative, two-stage stratified random sampling design and collected data through structured interviews using validated instruments. These methodological details, including sampling, data collection, and weighting procedures, have been comprehensively described in the NHMS 2019 Technical Report, which is appropriately cited in the manuscript (and URL has been included in the reference no 7). As for the criteria for data analysis, the selection of cases prior to data analysis was described under the subheading “Survey Respondents” on page 3 (line 117 – 126), and details of the type of statistical tests employed were detailed under subheading “Statistical Analyses” on page 5 (line 232-255). We believe this level of detail, together with the citation of the official report, ensures transparency and reproducibility while avoiding redundancy.
Comment 4: The results presented could be better organized and explained. Some sections lack a clear connection between the data presented and the conclusions drawn. It is important to reinforce the explanation of how the findings relate to the objective of the study.
Response 4: Thank you for the comment and suggestion. We have revised the table (Page 6, line 275-277, page 8, line 299-301), results (page 6-11, line 256-377) and conclusion (page 14, line 512-522) accordingly to provide better clarity or connection between the data presented.
Comment 5: The discussion section does not sufficiently explore the relationship between the study findings and the existing literature. It is necessary to include more relevant references and compare the results obtained with similar studies to strengthen the validity of the conclusions.
Response 5: Thank you for your valuable comment. We have revised the discussion section to include a more comprehensive comparison of our findings with existing literature. Specifically, we have incorporated relevant studies on sugar-sweetened beverage (SSB) consumption from Thailand and Indonesia, as well as findings from large national datasets including the U.S. National Health and Nutrition Examination Survey (NHANES), the China Health and Nutrition Survey (CHNS), and the Korean National Health and Nutrition Examination Survey (KNHANES). These additions provide important regional and global context to our results, allowing for a more robust interpretation and supporting the validity of our conclusions (Page 11-14)

Reviewer 3 Report
Comments and Suggestions for Authors
This study addresses the issue of sugar and sugary drinks consumption among the Malaysian population. The need to study the causes of non-communicable diseases is an important factor for their control, prevention and treatment.
The study examined the prevalence of daily sugar-sweetened beverages (SSB) consumption among Malaysian adults and its impact on public health. The results showed that SSB consumption is significantly more common among certain population groups.
Given that the study was conducted over a period of several months (and no significant association was found between SSB consumption and the risk of diabetes, hypertension, undiagnosed hyperlipoproteinemia or obesity), the results only reveal the high prevalence of SSB consumption among certain population groups.
Given that the study emphasizes the daily consumption of sugar-sweetened beverages (SSB) among adults, the title should be added and the need for integrated public health strategies should be emphasized.
For exemple -Daily Sugar-Sweetened Beverages Intake and its Association with Undiagnosed Non-communicable Diseases Among Malaysian Adults and the Need for Integrated Public Health Strategies: Findings from a Nationally Representative Cross-Sectional Study
Author Response
For Research Article: nutrients-3565763
Response to Reviewer 3’s comments
- Summary
We thank you for taking your time to review this manuscript. The comments/suggestions are relevant. Please find detailed responses below and the corresponding revisions/ corrections highlighted/ in track changes in the re-submitted files.
|
2. Questions for General Evaluation |
Reviewer’s Evaluation |
Response and Revisions |
|
Does the introduction provide sufficient background and include all relevant references? |
Yes |
We thank the reviewer for the acknowledgement |
|
Is the research design appropriate? |
Yes |
We thank the reviewer for the acknowledgement |
|
Are the methods adequately described? |
Yes |
We thank the reviewer for the acknowledgement |
|
Are the results clearly presented? |
Yes |
We thank the reviewer for the acknowledgement |
|
Are the conclusions supported by the results? |
Yes |
We thank the reviewer for the acknowledgement |
- Point-by-point response to Comments and Suggestions for Authors
Comment 1: This study addresses the issue of sugar and sugary drinks consumption among the Malaysian population. The need to study the causes of non-communicable diseases is an important factor for their control, prevention and treatment.
Response 1: We thank the reviewer for recognizing the importance of studying sugar and sugar-sweetened beverage consumption in the context of non-communicable disease (NCD) prevention and control. We appreciate the acknowledgement of our study’s relevance to public health efforts in addressing the growing burden of NCDs in Malaysia.
Comment 2: The study examined the prevalence of daily sugar-sweetened beverages (SSB) consumption among Malaysian adults and its impact on public health. The results showed that SSB consumption is significantly more common among certain population groups.
Response 2: We thank the reviewer for their thoughtful comment. We appreciate the recognition of our study's findings, particularly the observed disparities in daily sugar-sweetened beverage (SSB) consumption among specific population subgroups. These findings underscore the importance of targeted public health interventions to reduce SSB intake and mitigate its impact on non-communicable disease risk.
Comment 3: Given that the study was conducted over a period of several months (and no significant association was found between SSB consumption and the risk of diabetes, hypertension, undiagnosed hyperlipoproteinemia or obesity), the results only reveal the high prevalence of SSB consumption among certain population groups.
Response 3: We thank the reviewer for this insightful comment. We agree that, due to the cross-sectional design of the study, causal relationships cannot be established. Our aim was not to determine the direct association between SSB consumption and specific health outcomes, but rather to describe the prevalence of daily SSB intake among Malaysian adults with morbidities and to identify key sociodemographic patterns. We have clarified this point in the Discussion section and acknowledged the limitation of the study design in drawing causal inferences. We believe these findings still provide valuable insights for informing targeted public health strategies aimed at reducing SSB consumption in high-risk groups.
Comment 4: Given that the study emphasizes the daily consumption of sugar-sweetened beverages (SSB) among adults, the title should be added and the need for integrated public health strategies should be emphasized.
For exemple -Daily Sugar-Sweetened Beverages Intake and its Association with Undiagnosed Non-communicable Diseases Among Malaysian Adults and the Need for Integrated Public Health Strategies: Findings from a Nationally Representative Cross-Sectional Study
Response 4: We thank the reviewer for the suggestion. We will remain the title unchanged as the current manuscript focuses on the association between daily SSB intake and risk of non-communicable diseases. We, however, have included suggestions on the public health strategies in the discussion on curbing the high prevalence of SSB intake (page 12, line 368-382).

Round 2
Reviewer 1 Report
Comments and Suggestions for Authors
Yes, the manuscript has been improved than previous version.
Comments on the Quality of English LanguageOk
Author Response
comment: Yes, the manuscript has been improved than previous version.
response: We sincerely thank the reviewer for acknowledging the improvements made in the revised manuscript. We appreciate your constructive feedback, which helped us enhance the clarity and quality of our work.